# Vitamin D Influences the Activity of Mast Cells in Allergic Manifestations and Potentiates Their Effector Functions against Pathogens

**DOI:** 10.3390/cells12182271

**Published:** 2023-09-14

**Authors:** Yeganeh Mehrani, Solmaz Morovati, Sophie Tieu, Negar Karimi, Helia Javadi, Sierra Vanderkamp, Soroush Sarmadi, Tahmineh Tajik, Julia E. Kakish, Byram W. Bridle, Khalil Karimi

**Affiliations:** 1Department of Pathobiology, Ontario Veterinary College, University of Guelph, Guelph, ON N1G 2W1, Canada; ymehrani@uoguelph.ca (Y.M.); stieu@uoguelph.ca (S.T.); vanderka@uoguelph.ca (S.V.); jkakish@uoguelph.ca (J.E.K.); 2Department of Clinical Sciences, School of Veterinary Medicine, Ferdowsi University of Mashhad, Mashhad 91779-48974, Iran; n.karimi@mail.um.ac.ir; 3Division of Biotechnology, Department of Pathobiology, School of Veterinary Medicine, Shiraz University, Shiraz 71557-13876, Iran; s.morovati@shirazu.ac.ir; 4Department of Medical Sciences, Schulich School of Medicine & Dentistry, University of Western Ontario, London, ON N6A 3K7, Canada; hjavadi@uwo.ca; 5Department of Microbiology and Immunology, Faculty of Veterinary Medicine, University of Tehran, Tehran 14174-66191, Iran; sarmadisoroush@outlook.com; 6Department of Pathobiology, School of Veterinary Medicine, Ferdowsi University of Mashhad, Mashhad 91779-48974, Iran; ta.tajik@mail.um.ac.ir

**Keywords:** vitamin D, MC, vitamin D receptor, viral infection

## Abstract

Mast cells (MCs) are abundant at sites exposed to the external environment and pathogens. Local activation of these cells, either directly via pathogen recognition or indirectly via interaction with other activated immune cells and results in the release of pre-stored mediators in MC granules. The release of these pre-stored mediators helps to enhance pathogen clearance. While MCs are well known for their protective role against parasites, there is also significant evidence in the literature demonstrating their ability to respond to viral, bacterial, and fungal infections. Vitamin D is a fat-soluble vitamin and hormone that plays a vital role in regulating calcium and phosphorus metabolism to maintain skeletal homeostasis. Emerging evidence suggests that vitamin D also has immunomodulatory properties on both the innate and adaptive immune systems, making it a critical regulator of immune homeostasis. Vitamin D binds to its receptor, called the vitamin D receptor (VDR), which is present in almost all immune system cells. The literature suggests that a vitamin D deficiency can activate MCs, and vitamin D is necessary for MC stabilization. This manuscript explores the potential of vitamin D to regulate MC activity and combat pathogens, with a focus on its ability to fight viruses.

## 1. Vitamin D and Immunomodulation

Vitamin D is a fat-soluble secosteroid primarily produced in the skin and can also be consumed via food such as mushrooms, egg yolk, and oily fish. Vitamin D’s primary physiological function is metabolic homeostasis and regulation of calcium and phosphate metabolism, as well as the immunomodulatory effects that have gained attention through research [1,2,3]. 

The VDR is an essential nuclear receptor [4], responsible for forming the active compound of vitamin D, 1,25(OH)2D3 [5], and is an important signaling molecule, binding to promoter regions, regulating the transcriptional activity of target genes, and mediating the downstream biological effects of active vitamin D [5]. The receptor is present in immune cells such as neutrophils, MCs, macrophages (MQ), activated T and B cells, as well as innate lymphoid cells (ILCs) [6,7]. This wide distribution of the VDR in the immune system highlights its significance in the immune function. By interacting with the VDR, vitamin D helps maintain the immune balance, influences the immune response, and prevents excessive inflammation [8,9].

1,25(OH)2D3, a biologically active form of vitamin D, uses two distinct signaling pathways. The first pathway is the genomic pathway, in which 1,25(OH)2D3 binds to the VDR, thereby regulating transcription. The non-genomic pathway involves plasma membrane-associated receptors. This pathway causes immediate cellular responses without requiring transcriptional regulation [10]. By binding with the VDR, a member of the nuclear hormone receptors’ superfamily which is located in most human cells, including the cells from the immune system, vitamin D can act as a ligand-inducible transcription factor [11]. Moreover, immune cells such as MCs and MQs express the enzyme 1-α-hydroxylase (CYP27B1), which can convert 25(OH)D2 to the bioactive form of vitamin D (calcitriol). The ability of immune cells to synthesize calcitriol allows them to quickly increase vitamin D levels locally, inducing immune tolerance and immunomodulatory effects [12,13]. Vitamin D increases the number of regulatory T cells (Tregs), which have the ability to suppress harmful immune responses and reduce disease severity in many autoimmune diseases [14,15]. Vitamin D deficiency has been linked to the development of several autoimmune diseases, such as rheumatoid arthritis, inflammatory bowel disease, systemic lupus erythematosus, systemic sclerosis, multiple sclerosis, and type 1 diabetes [16]. In these cases, maintaining an adequate level of vitamin D is crucial to regulating the immune response and reducing the severity of the disease [16]. In addition, vitamin D boosts the body’s natural immunity, which is its initial defense against harmful pathogens. This is achieved by promoting the production of antimicrobial peptides, including cathelicidins and defensins, that help to protect against infections [17,18]. Moreover, vitamin D enhances the ability of MQs to attract and engulf harmful substances [19].

There has been a suggestion that monocytes exhibit a high level of VDR expression, which renders these cells more susceptible to the differentiating effects of vitamin D. This phenomenon potentially facilitates a rapid autocrine mechanism for the subsequent maturation of monocytes into MQs. As a result, vitamin D deficiency impairs MQ maturation and the production of specific surface antigens by downregulating the membrane expression of major histocompatibility complex class II (MHC-II) molecules [13].

However, vitamin D also modulates adaptive immune functions by promoting an appropriate immune response to foreign antigens while ensuring that the response does not become excessive [18]. One important finding was that there was a significant increase in the number of NK cells and Natural killer T (NKT) cells following vitamin D supplementation. Patients with disorders impacting vitamin D’s metabolism, such as chronic renal failure and vitamin D-resistant rickets, have been found to have decreased NK cell activity. In these instances, NK cell activity improved and even returned to normal as a result of the vitamin D supplementation [20]. Studies have identified novel MC-derived molecules, in either a secreted or membrane-bound form, facilitate communication between various immune cells [21], such as T cells [22], B cells [23], NK cells [24], innate lymphoid cells [25], eosinophils [26], MQs [27], and DCs [28]. The impact of vitamin D levels on these intercellular interactions is currently unknown and requires further research. Vitamin D, beyond its role in calcium homeostasis and bone health, plays a significant role in immune tolerance and possesses immunomodulatory effects.

## 2. Vitamin D and IgE-Mediated MC Activation

MCs are derived from hematopoietic stem cells. These cells leave the bone marrow and circulate in an undifferentiated state throughout the body, eventually settling in peripheral tissues like the skin, respiratory mucosa, and gastrointestinal tract [29]. MCs are pro-inflammatory effector cells that are critically involved in the pathogenesis of IgE-dependent inflammatory and allergic reactions, including anaphylaxis, allergic rhinitis, atopic dermatitis, and allergic asthma [30,31]. Upon activation of FcεRI following the binding of the specific antigen to IgE and the subsequent cross-linking of IgE molecules, a diverse array of preformed granule-associated mediators (histamine, serotonin) and newly synthesized pro-inflammatory mediators, cytokines, and chemokines are released [32]. Under normal circumstances, IgE-mediated MC activation is beneficial in eliminating invading pathogens. However, if unregulated, IgE-mediated MC activation can result in excessive inflammation and tissue damage [32]. Activation of MCs occurs in two separate phases: immediate degranulation and delayed secretion [33]. In the first phase of activation, MCs release their granules containing various pre-synthesized proteases, chemokines, and cytokines. In the late phase, they release de novo synthesized inflammatory mediators such as cytokines, growth factors, and lipid mediators (leukotrienes (LT) and prostaglandins [34]) over a span of 4–6 h after activation [35]. The curtailment of the release of these vasodilatory and pro-inflammatory mediators by MCs requires the presence of functional VDRs [31]. In addition to inhibiting IgE-mediated MC activation, vitamin D also plays a crucial role in MC stability [30]. Yip et al. demonstrated that calcitriol was able to suppress IgE-induced MC-derived pro-inflammatory and vasodilatory mediator production in vitro in a VDR-dependent manner. Bone marrow-derived cultured MCs (BMCMC) exposed to 1α,25(OH)2D3 for 24 h following IgE + specific antigen-induced activation of BMCMCs resulted in reduced IgE-mediated histamine release, as well as a reduction in cysteinyl leukotriene, TNF, and IL-6 production [31]. According to Liu et al., MCs were observed to be automatically activated in a vitamin D-deficient environment; however, in the presence of vitamin D, IgE-sensitized MC activation was inhibited. The activation of MCs in the context of inflammatory disorders is undesirable as it triggers MC degranulation resulting in the release of pro-inflammatory mediators and, thus, inflammation [36]. This information indicates that vitamin D is essential for modulating MC activation and maintaining MC stabilization, whereas the lack of vitamin D provokes MC stimulation.

## 3. Vitamin D and MCs in Allergy

Uncontrolled MC activity following the Th2 immune response is linked to the development of IgE-dependent allergic disorders. IgE antibodies cross-link to high-affinity IgE receptors during an allergic response and create IgE/FcεRI complexes on MCs, causing sensitization. When the body is re-exposed to the same antigens, they attach to the IgE/FcεRI complexes, serving as the primary immunological trigger for MC activation. As a result, degranulation and the release of proinflammatory mediators by the MCs start allergic reactions in the forms of type I hypersensitivity reactions and anaphylactic reactions [37,38]. In recent years, efforts have been made to find prospective therapeutic candidates that could potentially treat allergic reactions by limiting the activation of MCs [39,40,41]. Many factors are known to play a role in the emergence of these illnesses. However, there is rising concern regarding a potential connection with vitamin D deficiency. Vitamin D can have implications beyond bone metabolism and maintaining the calcium balance in the body, extending to immune-related conditions [42,43]. Although vitamin D is known to have a favorable impact on illnesses linked to an overactive Th1 immune response, its effects are more complicated in the contexts of allergic diseases (e.g., allergic asthma, allergic rhinitis, and eczema), where Th2 cells play a prominent role [44]. Infants with eczema within the first year of life are more likely to have inadequate levels of maternal 25OHD3 during pregnancy [45]. Additionally, studies have shown that children with asthma with higher serum IgE levels also have lower serum vitamin D levels [12]. The precise inhibitory mechanism on MCs remains to be fully elucidated. However, increasing evidence supports the pivotal role of vitamin D in the maturation, differentiation, homing, and stability of MCs within the body (Figure 1).

Some scientists have demonstrated the inhibitory impact of 1,25(OH)2D3 on the Th2 response, which is a potent stimulator of MC activation [29,46,47]. In an asthma model, 1,25(OH)2D3 dramatically lowered the inflammatory responses in the airways by preventing naive T cells from polarizing towards Th2 and by decreasing the levels of IL-4 in the bronchoalveolar lavage (BAL) fluid. Along with modulating cytokine transcription, 1,25(OH)2D3 also controls cell homing to lymph nodes, effectively suppressing the inflammatory response [46]. 

MCs rely on the presence of vitamin D to maintain their stability, both in their inactive state and during the sensitization [48]. Liu et al. reported that MCs are spontaneously activated in a vitamin D-deficient environment. To investigate this, they exposed MCs to calcitriol, the active form of vitamin D, which resulted in the upregulation of the VDR in these cells. Subsequently, signaling pathways related to the MC activation were suppressed following VDR attachment to the Src family kinase (SFK) Lck/Yes novel tyrosine kinase. Lyn was unable to join the FcεRI and MyD88 chain during this process. This led to decreased levels of spleen tyrosine kinase (Syk) phosphorylation, MAPK, and nuclear factor kappa light chain enhancer of activated B cells (NF-κB), and TNF-α transcription in MCs. The decrease was caused by the acetylation of RNA polymerase II and histone H3/H4. It may be helpful to consider the role of vitamin D’s metabolism in MC-related diseases to develop effective treatment approaches [30]. 

MCs play a significant role in the emergence of chronic spontaneous urticaria (CSU), another allergic disorder characterized by itchy, raised skin welts. Antihistamines do not work effectively for many CSU patients, and extended systemic steroid use is not the best course of action. By inhibiting the PI3K/Akt/p38 MAPK/HIF-1 signaling cascade axis, Zhao et al. found that vitamin D therapy may help people with CSU by reducing the expression of VEGF in MCs. They also suggested that vitamin D binding protein (VDBP), which transports vitamin D in the human body, may serve as a biomarker for detecting CSU 12 [49].

While it was previously believed that MCs exclusively contributed to allergic reactions brought on by innate and environmental triggers [50,51], Biggs, et al. showed that these cells may release IL-10, which may minimize the skin damage caused by prolonged UVB exposure. The precise process causing MCs to release the anti-inflammatory cytokine IL-10 is still unknown. However, 1,25(OH)2D3 production rises when the skin is exposed to UVB radiation. In turn, 1,25(OH)2D3 increases the mRNA level and release of IL-10 in cutaneous MCs mediated by VDRs. This process happens without leading to MC degranulation, which can reduce skin-related issues and inflammation caused by prolonged exposure to UVB radiation [52].

Overall, the prevalence of allergy illnesses has substantially increased in recent years. There are undoubtedly many factors that contribute to the emergence of these conditions, but low vitamin D levels may be more related. However, the balance and control of MCs may be preserved by having adequate levels of vitamin D. In this context, vitamin D supplementation or specific agonists that activate VDRs may be employed to treat allergic reactions triggered by MCs.

## 4. Vitamin D and Host MCs Defence against Pathogens

MCs actively contribute to the immune responses against pathogens, including bacteria, fungi, parasites, and viruses. For instance, MCs act against bacteria either directly by antimicrobial mechanisms or indirectly by recruiting or modulating the function of other immune cells [53]. Studies by Carlos et al. showed that adoptive transfer of wild-type MCs into TLR2-deficient mice restored the ability of these mice to limit airway infection with *Mycobacterium tuberculosis* (Mtb) and augmented the recruitment of myeloid cells effectively in the granuloma formation [54]. Interestingly, the presence of vitamin D proved to play a role in the immune responses against *Mycobacterium tuberculosis* by activating cathelicidin. Also, activation of TLR2/1 led to antimycobacterial activities that were dependent on vitamin D [55]. 

It has been shown that MCs can react to *Candida albicans* (*C. albicans*), which may be associated with vitamin D. This includes processes such as degranulation, recruiting neutrophils, decreasing fungal viability, and releasing anti-inflammatory substances [56]. It has been discovered that stromal MCs have the ability to kill phagocytosed yeast, while mucosal MCs cannot. In addition, when confronting fungal hyphae, stromal MCs produce higher concentrations of transforming growth factor (TGF)-β and IL-10 than mucosal MCs, leading to increased mucosal immune tolerance [57]. It was observed that vitamin D increased the expression of TGFβ [58]. This implies that vitamin D may influence the production of TGF-β by MCs.

The relationship between viral infections and vitamin D is fascinating, considering it is more sophisticated and complex than previously believed. The stimulation of an anti-viral environment, interactions with cellular and viral components, incitement of apoptosis and autophagy, as well as genetic and epigenetic changes, are only a few of the impacts that have been observed [59]. Hence, the review in Section 5 provides an overview of vitamin D’s impact on MC-mediated antiviral responses.

## 5. Vitamin D Affects MCs in Fighting Viral Infection and Viral Diseases

The link between vitamin D deficiency and susceptibility to viral infections has long been suggested, as vitamin D levels correspond to the severity of the immune response against viral infection [60]. Epidemiologic studies over the past two decades have shown a strong link between seasonal variations in vitamin D levels and a higher incidence of infectious diseases such as septic shock, respiratory infections, and influenza. Previous studies have also indicated that the severity of the immune response to viral infections is linked to vitamin D levels [61,62,63].

Vitamin D has direct antiviral effects (Figure 2) that are particularly effective against enveloped viruses, including SARS-CoV-2, the influenza virus, Human Immunodeficiency Virus (HIV), EBV, and the Hepatitis C Virus (HCV) [64]. Children with nutritional rickets were more susceptible to respiratory tract infections, indicating a link between the two [65]. There is clinical evidence to support calcitriol-induced immune protection against several viral pathogens, including HIV, hepatitis viruses, SARS-CoV-2, and the influenza virus, and 1,25(OH)2D has been reported to reduce susceptibility to HIV infection by preventing viral entry, lowering surface CD4 expression, and restricting monocyte proliferation. Indeed, it has been shown that vitamin D modulates TLR activity, receptors that identify microbes and express on immune cells, such as MQ, neutrophils, DCs, NK cells, basophils, eosinophils, and MCs [60]. Spirig et al. discovered that introducing vitamin D increased the production of endothelin-1 by DCs in response to TLR2 and TLR4 agonists [66]. The presence of vitamin D increased the responsiveness of monocytes to TLR2, 3, and 4 agonists, leading to the production of IL-8, a vital cytokine in fighting infections [67]. Research has shown that vitamin D activation in human corneal epithelial cells (HCEC) during TLR-induced inflammation has a dual effect. Firstly, it suppresses the expression of inflammatory mediators that are activated by TLRs. Secondly, it increases the production of antimicrobial peptides. This implies that vitamin D not only reduces the inflammatory response triggered by TLR stimulation but also aids in immune defense by promoting the production of antimicrobial peptides in human corneal epithelial cells [68].

Individuals with inflammatory disorders rely on TLRs in their cellular populations to stimulate the innate immune response. Monocytes and MQs are known to express significant amounts of TLR2 and TLR4. TLRs are crucial in activating the immune system to protect the body. However, it is essential to note that excessive inflammation caused by TLRs can harm the host, such as sepsis or autoimmune disorders. In a recent study conducted by Choi et al., it was found that Vitamin D can reduce inflammation resembling Behçet’s disease, induced by the herpes simplex virus, in a mouse model. This effect is achieved by lowering TLR2 and TLR4 expression in monocytes [69].

The research on how vitamin D affects MCs during viral infections is still ongoing. MCs release inflammatory cytokines such as IL-1, IL-6, and IL-33, as well as substances like histamine and proteases in response to viral stimulation [70]. Vitamin D can influence the behavior of MCs, causing them to release anti-inflammatory cytokine IL-10 instead of pro-inflammatory mediators. As a result, the production of IFN-γ, IL-2, IL-3, TNF-α, and GM-CSF is suppressed by IL-10. This helps to prevent an excessive immune response that may be caused by the p38 MAPK/NF-κB signaling pathways [52,71].

There is increasing evidence that vitamin D supplementation activates both innate and adaptive immune cells, which can help alleviate pathogenic inflammation caused by viral infections in MCs. Calcitriol promotes a more balanced response to viral infections by shifting from the pro-inflammatory Th1 and Th17 to the anti-inflammatory Th2 and Treg profiles [72]. In fact, inflammation is regulated by vitamin D, which inhibits Th1 and Th17 cells and their cytokines while promoting anti-inflammatory cytokines, resulting in minimized organ damage [73].

Viral infections involve complex immune responses, such as cross-talk between T cells and MCs. They can influence each other and regulate immune responses in different ways. T cell-dependent mediators like β-chemokines incite MC degranulation, while MC-derived cytokines like IL-16 or IL-4 promote T-cell migration and differentiation toward Th2 subpopulations [72,73,74]. Therefore, a deficiency in vitamin D may hinder the ability of immune cells to fight off viruses.

Evidence showed that a lack of vitamin D may decrease the immune system’s ability to create activated T lymphocytes, particularly CD8^+^ T cells. These cells are essential in targeting virally infected B cells and suppressing viral infections, such as the EBV [75,76]. During viral infections, B cells play a crucial role in secreting IgG antibodies specific to viral antigens. These IgGs then bind to the MCs through their Fc region, triggering MC activation [26]. It has been suggested that vitamin D inhibits the maturation and switching of B-lymphocytes [77]. Although vitamin D may temporarily affect the production of antibodies that combat viral infections, this effect can diminish over time. Røsjø et al. demonstrated that prolonged high-dose vitamin D consumption has been linked to decreased IgG levels against the EBNA-1, but only during a 48-week period [78].

NK cell activity and total serum IgE levels have been found to be positively associated with each other [74]. Activated NK cells triggered by IgE can produce cytokines and chemokines, such as IFN-γ, TNF-α, granulocyte-macrophage colony-stimulating factor (GM-CSF), MIP-1α, MIP-1γ, and TGF-B, and are capable of performing cytotoxicity against IgE-coated cells through a FcƳRIII-dependent method [79]. The effect of vitamin D on NK cells in COVID-19 patients has not yet been studied. Earlier research suggests that those with chronic conditions may have lower blood calcitriol levels, which could reduce NK cell cytotoxicity. Therefore, administering vitamin D may improve the NK cell function, which is crucial for the immune response to infections like SARS-CoV-2 [80]. The emerging evidence increasingly supports the pivotal role of MCs in recruiting NK cells to infection sites. For instance, John et al. found that MCs stimulated the trafficking of NK cells into contaminated skin to restrict the Dengue Virus (DENV) [81]. DENV is typically transmitted through the bite of an infected mosquito, which then introduces DENV particles into the skin. Type I interferon, particularly IFNα, plays a pivotal role in the body’s defense against the DENV. It functions to suppress viral replication and also aids in the activation of cells capable of eliminating DENV-infected cells, such as T cells and NK cells. Nonetheless, the specific cell types that initially detect the DENV are not entirely understood. Given their prevalence in the skin, it is plausible that MCs could be among the first to come into contact with the DENV. In more detail, it has been observed in laboratory studies that certain cellular sensors responsible for detecting viral RNA, including TLR3, MDA5, and RIG-I, play a crucial role in initiating an anti-DENV pathway. This transcriptional process aimed at combating the DENV infection involves the activation of the type I interferon response and the generation of chemokines such as CXCL12 and CX3CL1. These chemokines facilitate the recruitment of NK cells, which possess the potential to eliminate cells infected by the DENV [53]. 

MCs are not susceptible to direct infection by the RSV. Instead, when co-cultured with RSV-infected airway epithelial cells, they demonstrated a secondary reaction to the virus. Due to their extended lifespan and residence within the respiratory tract, MCs have the ability to act as sentinel cells, promptly detecting and initiating a coordinated immune response to the RSV infection. In individuals with the RSV infection, the main physiological function of MCs is believed to be the promotion of effector cell recruitment through the generation of mediators such as IFN-type I, CCL4, and CXCL10. According to a study conducted by Al-Afif et al., MCs possess the capability to produce notable quantities of CCL4, CCL5, and CXCL10, which are known to be involved in the recruitment of NKT cells, T-cells, and monocytes. However, MCs do not produce significant levels of CCL11 or CXCL8, which are primarily responsible for eosinophil and neutrophil recruitment. This observation indicates that MCs play a role in curbing RSV rather than causing detrimental bronchiolitis [82]. 

In two studies, researchers examined the selective recruitment of NK cells by human cord blood-derived MCs. The results indicated that human NK cells migrated to infected tissue through the CXCL8 and CXCR1 axis in MCs when triggered by the RSV or a virus-associated TLR3 agonist [45,83]. Some of the most significant chemicals released by infected MCs include IL-10, type I, and type III IFNs. These contribute to the amplification of IFN-γ secretion by NK cells [84].

MCs can undergo productive infection by the reovirus, which triggers a robust immune response involving the expression of all 12 members of the IFNα family as well as IFN-β. These IFNs also have an autocrine effect on MCs, increasing the production of CXCL10, as well as VEGF and IL-1RA, which supports tissue remodeling and reduce inflammation. Moreover, NK cells can be attracted and further triggered by the substantial quantities of type I IFNs released by infected MCs, which subsequently results in the production of IFN-γ and facilitates the activation of other immune effector cells [84].

MQs as phagocytic cells are essential for engulfing and destroying infected cells and viral particles. MCs can be recruited and activated by these immune cells through the secretion of pro-inflammatory cytokines such as TNF-α and IL-1. The activation of TLR1/2 on MQs by viral components leads to the expression of CYP27B1 and VDR, which are involved in vitamin D metabolism and signaling. Vitamin D promotes the differentiation of MQs. Additionally, it increases the expression of CD14, a pathogen recognition molecule, and TLRs 2 and 4 in MQs [85]. The available data on the interaction between VDR and viral proteins and cellular transcription factors indicate that vitamin D may play a role in viral infections.

Discovering VDR expression in MC and its vitamin D responsiveness created a new regulatory axis between vitamin D and this innate immune cell. Nonetheless, the mechanisms by which vitamin D exerts its potential remain largely undetermined [86]. Here, we discuss how vitamin D affects cytokine and antimicrobial peptide generation by MCs, which combat pathogens through PRR activity.

MCs recognize pathogens through various PRRs, including TLRs, NOD-like receptors (NLRs), RIG-like receptors (RLRs), and Dectin-1 [87]. TLRs play a significant role in the initial inflammatory response to viral infection. However, there is limited research on the impact of vitamin D intake on viral infections via MC TLRs. The interaction between the VDR and various transcription factors is important for combating viruses. Although the precise mechanisms have not yet been established, pre-exposure to vitamin D has also been found to affect HSV-1 immunopathogenesis by reducing the expression of TLR2 and 9 [88]. A study by Wang et al., revealed that vitamin D induces the expression of pattern recognition receptor NOD2/CARD15/IBD1 gene and protein in cells of monocytic and epithelial origin through two distal VDREs located in the NOD2 gene, which finally result in the induction of DEFB4 and CAMP production [89]. DEFB4 and CAMP can influence an MC-dependent immune responses. The two major classes of antimicrobial peptides (AMPs) in mammals against Gram-positive and Gram-negative bacteria, enveloped viruses, and fungi are defensins (defensin Beta 4 (DEFB2/DEFB4)/human Beta defensin 2 (HBD2)) and cathelicidin (CAMP/LL37) [90]. Both humans and mice MCs express cathelicidin, but there are limited in vivo studies on their antimicrobial functions [91]. It has been demonstrated that cathelicidin production in MCs can be stimulated through TLR2 signaling in response to lipoteichoic acid (LTA) of commensal bacteria [92]. In addition, TLR2 activation positively affects MC recruitment to the site of the injury [92]. LL-37 (the C-terminal peptide of human cathelicidin antimicrobial peptide) can act as a chemotactic agent, stimulate degranulation in MCs, and positively affect MC responses to infection [83,93]. β-defensins and the α-defensin family are chemoattractive and activate MCs, similar to cathelicidins. However, there is no evidence of defensin peptide production by human MCs. Nevertheless, mouse β-defensin-4 (mBD-4) displays the highest homology to human β-defensin-2 [91] and [94].

Moreover, it has been shown that β-defensins and cathelicidin activate human MCs via the Mas-related gene X2(Mrg-x2), a G protein-coupled receptor [95,96]. LL-37 enhances TLR2, TLR4, and TLR9 on the MC surface and TLR3, TLR5, and TLR7 in the cell interior [97,98]. Furthermore, Agier et al. discovered that LL-37 and hBD-2 activate NOD1, NOD2, and RIG-I receptors and induce pro-inflammatory and migratory responses in murine peritoneal MCs [99]. Detection of the vitamin D response elements (VDREs) adjacent to the transcription start site of genes related to antimicrobial peptides implies that 1,25 D directly regulates the innate immune response against microbes [100]. Therefore, apart from the undetermined effect of vitamin D on MC cathelicidin production, this vitamin can lead to the robust induction of CAMP in other participating cells during immune response such as keratinocytes, monocytes, and neutrophils [101] and, therefore, may indirectly enhance the antimicrobial response of MCs by attracting them to the pathogen entry site, augmenting the MC capability and sensitivity to TLR, NLR, and RLR ligands, and inducing their degranulation and mediator release that may include MC CAMP itself. It has also been observed that vitamin D enhances the induction of HBD2, which may exert the same impact as cathelicidin on MCs [100].

MCs can play multiple roles in many pathological conditions through the plethora of mediators they can release. It has been discovered that they may be polarized through either pro-inflammatory (low expression of IL-10) or anti-inflammatory (high expression of IL-10) profiles [102]. For instance, MCs can limit the skin pathology induced by chronic ultraviolet (UV)-B irradiation via the production of IL-10. Surprisingly, 1α,25(OH)2D3 can up-regulate IL-10 mRNA expression and induce IL-10 secretion in MCs, thus mitigating the inflammation associated with chronic UVB exposure of the skin [52]. 

Another cytokine regulated by vitamin D is IL-33, which functions as an alarm signal and is typically present in stimulated epithelial cells like keratinocytes, pulmonary epithelial cells, and fibroblasts [103]. Its function relies on binding to the membrane-bound orphan IL-1 family receptor ST2 (Il1rl1) [104]. Il1rl1 also has a soluble form (sST2), which acts as a decoy receptor and sequesters free IL-33, preventing ST2/IL-33 signaling [105]. The membrane-bound form (ST2) activates the MyD88/NF-κB signaling pathway to enhance the innate lymphoid cell type 2, Th2, Treg, and MC functions [106]. In murine and human MCs, ST2/IL-33 signaling has been demonstrated to promote their survival, activation, and maturation, as well as induction of CCL17, CCL22, CCL2, CXCL8, IL-5, IL-13, and GM-CSF secretion [107,108,109]. Moreover, upon ST2 signaling, MCs can produce type 2 cytokines and may promote a Th17 response during airway inflammation [110,111]. Studies have shown that vitamin D deficiency can cause higher IL-33 levels, but taking vitamin D supplements can regulate IL-33 expression [112,113,114,115]. 

Additionally, vitamin D indirectly regulates Type I IFNs (α/β), another important family of cytokines. IFNs have been approved for treating certain viral infections [116]. The activation of interferon-regulatory factor 3 (IRF3) and subsequent production of interferon-β (IFN-β) in human embryonic kidney cell cultures is initiated by NOD2, leading to the induction of type I interferon antiviral responses. They also noted that NOD2-deficient mice have reduced interferon production and increased susceptibility to virus-induced pathogenesis [117]. As mentioned, LL-37 and hBD-2 have been shown to induce NOD2 expression in murine peritoneal MCs [99]. Considering the stimulatory effects of vitamin D on LL-37 and hBD-2 production [101], an indirect pathway from vitamin D to elevated production of type I interferon in MC seems plausible, although the validity of type I interferon production induced by NOD2 stimulation in MC is yet to be clarified.

## 6. Controversy in Roles That Vitamin D May Play in Viral Infection

Several studies have indicated a possible connection between vitamin D and the onset [59,118,119] and acceleration of viral diseases [59,120]. It has been shown that certain immune cells synthesize 1,25(OH)2D3 in response to infection [58]. According to Berkan-Kawińska et al. [121], even though those with chronic hepatitis C or chronic hepatitis B may have insufficient levels of 25(OH)D, there is no evidence that these deficiencies lead to a worse prognosis for patients. Additionally, studies have shown no link between vitamin D levels in blood and liver fibrosis [122,123] or poor responses to anti-viral treatments for HCV infection [124,125,126,127,128,129]. It is worth noting that the levels of 25(OH)D3 did not seem to affect the effectiveness of the treatment in the African-American population [129]. Additionally, studies have shown that there is no significant correlation between the vitamin D levels and liver inflammation in patients with the hepatitis B virus (HBV) infection [130,131,132,133]. However, previous studies that described the positive effects of vitamin D on HBV and HCV infections are contradicted by these findings [134,135,136,137,138,139].

Similar findings have been reported about respiratory viruses. There has been a plethora of studies suggesting that vitamin D affects the pathogenesis of influenza virus, parainfluenza virus, adenovirus, respiratory syncytial virus (RSV), and rhinovirus. For example, vitamin D can suppress the replication of rhinoviruses by triggering the expression of cathelicidin and IFN-stimulated genes [34,140,141,142]. Despite the considerable amount of evidence that demonstrates the beneficial role of vitamin D in reducing respiratory viral infections, other investigations have shown contradictory results. According to these investigations, treatment with vitamin D did not have any discernible effect on the severity or extent of respiratory viral infections, namely, influenza and rhinoviruses [143,144,145,146].

Numerous positive effects have also been attributed to vitamin D concerning HIV infection. Campbell et al. [147] revealed that 1,25(OH)2D3 hindered the replication of HIV by stimulating autophagy and promoting the maturation of phagosomes in PBMC infected by the HIVBa-L strain. Indeed, the amount of vitamin D and its corresponding receptor are associated with innate resistance against HIV-1 infection in isolated PBMCs [148]. In HIV patients undergoing antiretroviral therapy, insufficient levels of vitamin D were linked to a decrease in the CD4^+^ T-cell count [149]. However, some articles with exceptions have claimed contradicting results so that no remarkable associations were observed between HIV-related variables and levels of 25(OH)D3 in patients with HIV [149,150,151,152,153]. Also, it has been revealed that nutritional supplementation with vitamin D in HIV patients showed no improvements in mortality [150], HIV viral loads [154], and T cell subset counts [151,154].

A possible link between Epstein-Barr virus (EBV) infection and multiple sclerosis (MS) has been proposed [155,156,157]. There is a suggestion that EBV, along with other risk factors, may interact synergistically in susceptible individuals and lead to the development of MS. However, EBV by itself does not cause MS [158]. There is an ongoing debate about the association between vitamin D and anti-EBNA1 antibody levels in the bloodstream as a risk factor for MS [159].

This controversy may be due to differences like in vivo or in vitro studies, clinical trial design, utilized experimental systems, and established inclusion and exclusion criteria for study participants. However, these are a small number of plausible reasons for the controversial effects of vitamin D [160]. One of the interesting areas in vitamin D’s contribution to immune responses is the way it affects immune cells. MCs are well known for their important beneficial or detrimental role in immune responses to viral infections. Investigating the effect of vitamin D on MCs combatting viral infections may help to draw a more comprehensive picture of the potential preventive and therapeutic effects of vitamin D.

## 7. Summary and Conclusions

MC degranulation in SARS-CoV-2 infection due to numerous released granules in the alveolar spaces has been reported [161]. Vitamin D plays a crucial role in maintaining the stability of MCs, which can release histamine upon activation [162]. Moreover, some studies showed that compared to those with adequate vitamin D levels, individuals with a vitamin D deficit have a higher risk of contracting COVID-19, more severe symptoms, and a higher mortality rate [64,71,163]. 

Moreover, DENV infects MCs [53] and triggers both degranulation and cytokine production by MCs [81,164,165,166] and induces an inflammatory reaction via MCs that involves recruiting a variety of T cell subsets, NKT cells, and NK cells to the skin and lymph nodes [81,167]. Since severe dengue is linked to an imbalance in the production of proinflammatory cytokines, there has been a suggestion that vitamin D might exert regulatory control over pro-inflammatory cytokine levels through its interaction with signaling molecules involved in TLR activation. As a matter of fact, it has been reported that the CYP27B1-dependent generation of 1,25(OH)2D3, which in turn increases the expression of TLR3 and RIG-I, bolsters the type I IFN-dependent antiviral response [168]. The activation of MCs can occur due to exposure to the H5N1 virus. MC degranulation has been found in influenza-infected tissues [169,170]. This activation leads to the proliferation of MCs and the subsequent release of various mediators and cytokines, such as histamine, tryptase, and IL-6. In response to viral infections, MCs may initiate the cytokine storm by producing pro- and anti-inflammatory cytokines in the influenza infection [70,170]. It has been shown that vitamin D improves cellular immunity by reducing the cytokine storms caused by the innate immune system. Also, inducing antimicrobial peptides like LL-37, is one way that vitamin D improves cellular innate immunity. LL-37 inhibited the influenza A virus replication in the mouse model [171,172]. 

Vitamin D’s immunomodulatory properties affect both the innate and adaptive immune systems and play a crucial role in immune homeostasis. MCs are pro-inflammatory effector cells that play a crucial role in the pathogenesis of IgE-dependent inflammation. Our review compiles recent findings on the work done regarding the role that MCs play in the host’s defense against pathogens, as well as the immunomodulatory effects of vitamin D on the immune system, with special emphasis on MCs. Vitamin D stimulates an anti-viral environment, interacts with cellular and viral components, and induces genetic and epigenetic changes via its VDR. The presence of VDRs in MCs indicates that MC activity can be influenced by vitamin D. Attachment of vitamin D to VDR on MCs results in the upregulation of the inflammatory cytokine IL-10, which acts to suppress IL-1B, IL-2, IL-4, IL-6, and TNF-α production and ultimately promotes an anti-inflammatory environment. Also, it has been proven in the literature that vitamin D plays a crucial role in the maintenance of MC stabilization in both quiescent and sensitized conditions. Until recently, the importance of vitamin D in maintaining MC stability and its ability to suppress the production of MC vasodilatory and pro-inflammatory mediators were not known. In addition, it is worth exploring how vitamin D can enhance the function of MCs in combatting pathogens. This is due to the fact that MCs can produce cathelicidin, and vitamin D can induce and upregulate the production of this antimicrobial peptide, as well as β-defensin 2, in immune cells. Vitamin D also plays a role in influencing the expression of PRRs, which enable MCs to detect pathogens. Investigating these areas may provide insight into how vitamin D can improve disease outcomes, especially in patients who are deficient or have an insufficient amount of this vitamin.

## Figures and Tables

**Figure 1 cells-12-02271-f001:**
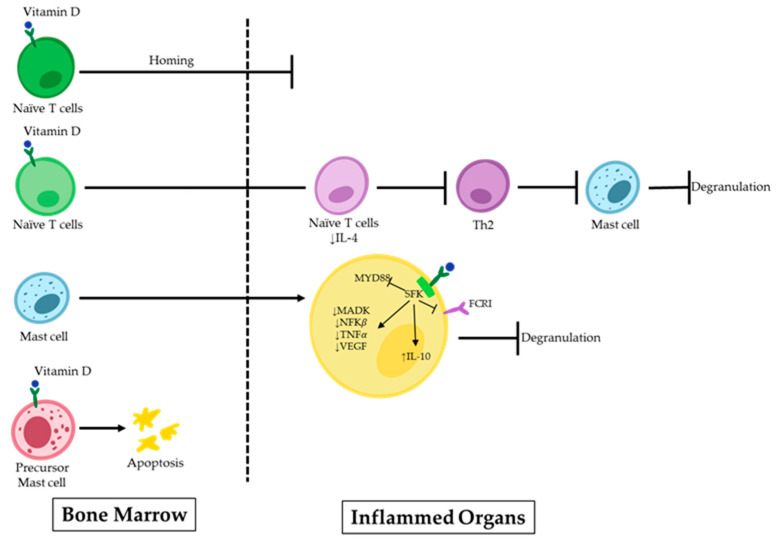
Vitamin D effects on MCs in allergic disorders. 1,25(OH)2D binds to the VDR on naïve T-cells, inhibiting their homing to inflamed organs. Additionally, it suppresses the polarization of these cells to Th2 and further MC degranulation. 1,25(OH)2D can induce apoptosis or suppress signaling pathways in precursor MCs upon direct binding.

**Figure 2 cells-12-02271-f002:**
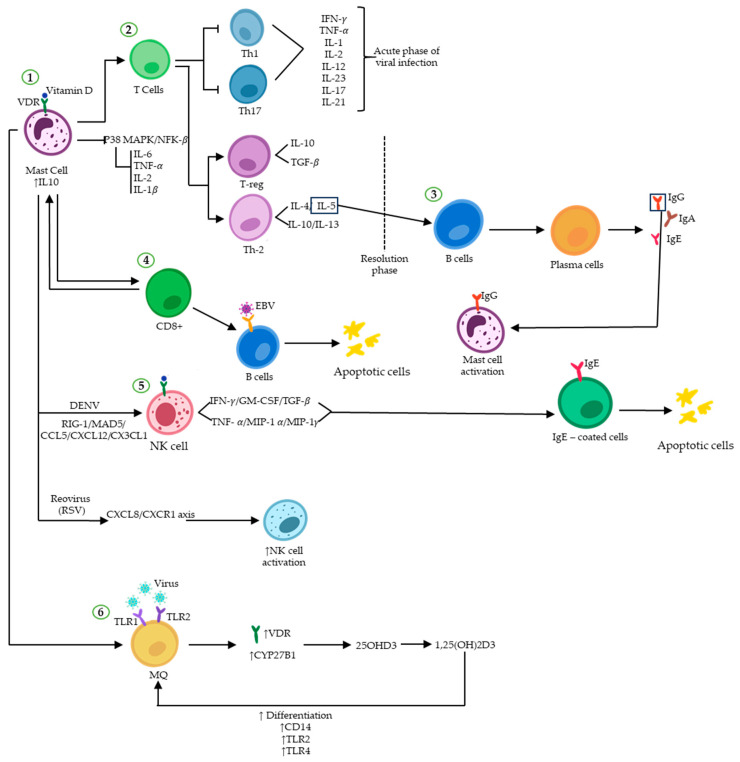
Vitamin D and immunoregulatory function during viral infections. Vitamin D binding to VDR induces and/or represses the transcription of many genes. (**1**) Attachment of vitamin D to VDR on MCs upregulates the expression of IL-10, ultimately promoting an anti-inflammatory environment by inhibiting the P38 MAPK/NF-κB pathway. (**2**) The immune response shifts from a Th1/Th17 pro-inflammatory profile to an anti-inflammatory Th2/Treg profile. (**3**) IL-5-activated plasma cells secrete different types of immunoglobulins. (**4**) The binary interplay between MCs and CD8^+^ T-cells results in the elimination of virus-infected B cells (such as EBV) through apoptosis. (**5**) NK cell activity can be enhanced by vitamin D in response to viral infections such as DENV and RSV, which trigger apoptosis in infected cells. (**6**) The expression of VDR and CYP27B1 increases in macrophages upon formation of the vitamin D-VDR dimer, resulting in enhanced immune response to viral attacks.

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
