# Peer review of "Vitamin D Influences the Activity of Mast Cells in Allergic Manifestations and Potentiates Their Effector Functions against Pathogens"

_cells, 2023, doi:10.3390/cells12182271_

Round 1
Reviewer 1 Report
The article by Mehrani et al. provided a review on summarizing the potential effect of vitamin D on Mast Cells in combating pathogens, the manuscript is overall interesting, however several points need to be addressed.
1) The title is inappropriate given the allergy part in the manuscript as allergy is usually not considered as pathogens, the title should be rephrased or the allergy part should be removed.
2) The sign of "Mast cell" is missing underneath the blue cell on the left.
3) In line 232, the "vitamin D levels are negatively correlated with outcome" is a bit misleading, please rephrase it to be more clear, is more vitamin D positively correlated with better outcome?
4)As the authors indicated, the vitamin D has immunomodulatory properties on both innate and adaptive immune systems, however in the innate immune system part, despite NK and Mac, is insufficiently discussed, and how this is connected to mast cell and vitamin D should be more clearly addressed.
5) The authors seem to attract an attention to vitamin D-Mast Cells-viral infection, as we know, type I interferon is critical in eliminating virus, however it is less discussed here, it is worth more effort in bringing more.
6) The figure legend needs to be specific to each item displayed like the yellow residues may represent viruses, however it seems not to be explained properly.
7) The transitions between IL-10, IL-33, NOD2 signaling is a bit jumpy, the flow needs refined.
Fine.
Author Response
"Please see the attachment."

Reviewer 2 Report
This review on the immunomodulatory effects of vitamin D on the immune system addresses a new point of view in reference to cells of innate and adaptive immunity, particularly regarding Mast Cells.
The manuscript is quite organized and described. The bibliographic data are very recent. However, I suggest a minor revision to refine the quality of the manuscript. Authors should check and correct some typos.
Minor revisions:
Line 31-33 The sentence could be written better.
Line 154-159 The period is unclear, the Authors should also arrange the punctuation better.
Line 210 The Authors should insert the bibliographic note or in any case specify which fungal infections they are dealing with.
Line 245-247 Perhaps the Authors could briefly explain how Vitamin D can modulate the activity of TLRs.
Line 453-459 Too long a period, I suggest splitting it into two or more sentences.
Author Response
"Please see the attachment."

Round 2
Reviewer 1 Report
Thanks the authors for their efforts addressing my comments.
Point 3, the authors still not give a clear conclusion on the link between vitamin D and viral infection or incidence of infectious diseases, as there is some controversy. If the link is not clear, the author should remove this part or phrase it very clearly. For now, it is very vague description and the authors should avoid that.
Fine.
Author Response
Point 3, the authors still not give a clear conclusion on the link between vitamin D and viral infection or incidence of infectious diseases, as there is some controversy. If the link is not clear, the author should remove this part or phrase it very clearly. For now, it is very vague description and the authors should avoid that.
Response: We thank the reviewer for evaluating our manuscript and providing valuable feedback. We have created a new section responding to the reviewer's concerns about the relationship between vitamin D and viral infections.
Controversy in roles that vitamin D may play in viral infection
Several studies have indicated a possible connection between vitamin D and the onset [59, 118, 119] and acceleration of viral diseases [59, 120]. It has been shown that certain immune cells synthesize 1,25(OH)2D3 in response to infection [58]. According to Berkan-Kawińska et al. [121], even though those with chronic hepatitis C or chronic hepatitis B may have insufficient levels of 25(OH)D, there is no evidence that these deficiencies lead to a worse prognosis for patients. Additionally, studies have shown no link between vitamin D levels in the blood and liver fibrosis [122, 123] or poor responses to anti-viral treatments for HCV infection [124-129]. It is worth noting that the levels of 25(OH)D3 did not seem to affect the effectiveness of the treatment in the African-American population [129]. Additionally, studies have shown that there is no significant correlation between vitamin D levels and liver inflammation in patients with hepatitis B virus (HBV) infection [130-133]. However, previous studies that described the positive effects of vitamin D on HBV and HCV infections are contradicted by these findings [134-139].
Similar findings have been reported about respiratory viruses. There has been a plethora of studies suggesting that vitamin D affects the pathogenesis of influenza virus, parainfluenza virus, adenovirus, respiratory syncytial virus (RSV), and rhinovirus. For example, vitamin D can suppress the replication of rhinoviruses by triggering the expression of cathelicidin and IFN-stimulated genes [34, 140-142]. Despite a considerable amount of evidence that demonstrates the beneficial role of vitamin D in reducing respiratory viral infections, however, other investigations have shown contradictory results. According to these investigations, treatment with vitamin D did not have any discernible effect on the severity or extent of respiratory viral infections, namely influenza and rhinoviruses [143-146].
Numerous positive effects have also been attributed to vitamin D concerning HIV infection. Campbell et al. [147] revealed that 1,25(OH)2D3 hindered the replication of HIV by stimulating autophagy and promoting the maturation of phagosomes in PBMC infected by the HIVBa-L strain. Indeed, the amount of vitamin D and its corresponding receptor are associated with innate resistance against HIV-1 infection in isolated PBMCs [148]. In HIV patients undergoing antiretroviral therapy, insufficient levels of vitamin D were linked to a decrease in the CD4+ T-cell count [149]. However, some articles with exceptions have claimed contradicting results so that no remarkable associations were observed between HIV-related variables and levels of 25(OH)D3 in patients with HIV [149-153]. Also, it has been revealed that nutritional supplementation with vitamin D in HIV patients showed no improvements in mortality [150], HIV viral loads [154], and T cell subset counts [151, 154].
A possible link between Epstein-Barr virus (EBV) infection and multiple sclerosis (MS) has been proposed [155-157]. There is a suggestion that EBV, along with other risk factors, may interact synergistically in susceptible individuals and lead to the development of MS. However, EBV by itself does not cause MS [158]. There is an ongoing debate about the association between vitamin D and anti-EBNA1 antibody levels in the bloodstream as a risk factor for MS [159].
This controversy may be due to differences like in vivo or in vitro studies, clinical trial design, utilized experimental systems, and established inclusion and exclusion criteria for study participants. However, these are a small number of plausible reasons for the controversial effects of vitamin D [160]. One of the interesting areas in vitamin D's contribution to immune responses is the way it affects immune cells. MCs are well known for their important beneficial or detrimental role in immune responses to viral infections. Investigating the effect of vitamin D on MCs combatting viral infections may help to draw a more comprehensive picture of the potential preventive and therapeutic effects of vitamin D.